# FiLM-Ensemble: Probabilistic Deep Learning via Feature-wise Linear Modulation

**Mehmet Ozgur Turkoglu**
ETH Zurich

**Alexander Becker**
ETH Zurich

**Hüseyin Anil Gündüz**
LMU Munich

**Mina Rezaei**
LMU Munich

**Bernd Bischl**
LMU Munich

**Rodrigo Caye Daudt**
ETH Zurich

**Stefano D'Aronco**
ETH Zurich

**Jan Dirk Wegner**
ETH Zurich &
University of Zurich

**Konrad Schindler**
ETH Zurich

## Abstract

The ability to estimate epistemic uncertainty is often crucial when deploying machine learning in the real world, but modern methods often produce overconfident, uncalibrated uncertainty predictions. A common approach to quantify epistemic uncertainty, usable across a wide class of prediction models, is to train a *model ensemble*. In a naïve implementation, the ensemble approach has high computational cost and high memory demand. This challenges in particular modern deep learning, where even a single deep network is already demanding in terms of compute and memory, and has given rise to a number of attempts to emulate the model ensemble without actually instantiating separate ensemble members. We introduce FiLM-Ensemble, a deep, implicit ensemble method based on the concept of Feature-wise Linear Modulation (FiLM). That technique was originally developed for multi-task learning, with the aim of decoupling different tasks. We show that the idea can be extended to uncertainty quantification: by modulating the network activations of a single deep network with FiLM, one obtains a model ensemble with high diversity, and consequently well-calibrated estimates of epistemic uncertainty, with low computational overhead in comparison. Empirically, FiLM-Ensemble outperforms other implicit ensemble methods, and it comes very close to the upper bound of an explicit ensemble of networks (sometimes even beating it), at a fraction of the memory cost.

## 1 Introduction

A key component for reliable and trustworthy machine learning are algorithms that output not only accurate predictions of the target variables, but also well-calibrated estimates of their uncertainty [Gal and Ghahramani, 2016]. The overall uncertainty of a predictor is usually decomposed into two parts [Der Kiureghian and Ditlevsen, 2009]. *Aleatoric uncertainty* is inherent in the data, for instance due to class overlap or sensor noise. On the contrary, *epistemic uncertainty* characterises the uncertainty of the model weights, due to a lack of knowledge about parts of the input space that are insufficiently represented in the training set. Uncertainty caused by distributional shifts between the training and test data is sometimes conceived as a third source of uncertainty [Malinin and Gales, 2018], but in practice often modelled as part of epistemic uncertainty.

Measuring epistemic uncertainty for complex models such as deep neural networks is not trivial: by definition one cannot derive it from the training data, since it concerns the behavior of the model

36th Conference on Neural Information Processing Systems (NeurIPS 2022).

in regions of the input space that are not represented in the training set. Different methods have been explored [Welling and Teh, 2011, Graves, 2011, Blundell et al., 2015, Lakshminarayanan et al., 2017a, Huang and Belongie, 2017, Gal and Ghahramani, 2016], however the de facto standard remain deep ensemble models [Lakshminarayanan et al., 2017a]. In its basic form, such a *deep ensemble* is simply a collection of independently trained networks that can be regarded as Monte Carlo samples from the model space. To obtain ensemble members with reasonably low correlation, one can exploit the stochastic nature of the optimisation procedure, with different (random) weight initialisation and different (random) batches during stochastic gradient descent. The expectation is that, once trained, the ensemble members will agree for inputs near the training samples, since the loss function favours similar outputs at those locations. Whereas they may disagree in unseen regions of the data space. Thus, the spread of their predictions is a measure of epistemic uncertainty. The ensemble idea is conceptually very simple, but nevertheless yields uncertainties that are well calibrated, i.e., they are in line with the actual prediction errors (in expectation).

The drawback of deep ensembles is their large computational cost. Both computation and memory consumption grow directly proportionally with the number of ensemble members, during training as well as during inference. This makes them impractical in hardware-constrained settings, and leads to a widening gap as models continue to grow in size, while at the same time applications like mobile robotics, augmented reality and smart sensor networks increase the need for mobile and embedded computing.

To improve efficiency, several researchers have explored ways to mimic deep ensembles without explicitly duplicating the underlying network. Possible strategies include the reuse and recombination of network modules [e.g., Wen et al., 2020, Havasi et al., 2021], injection of noise at inference time [e.g., Gal and Ghahramani, 2016], as well as hybrid variants [e.g., Durasov et al., 2021]. Empirically, these models do speed up training and/or inference, but still exhibit a significant performance gap compared to the naïve, explicit ensemble, both w.r.t. prediction quality and w.r.t. the calibration of the predicted (epistemic) uncertainties.

The lottery ticket hypothesis [Frankle and Carbin, 2018] and other network pruning studies, e.g., by Han et al. [2015], Lee et al. [2019], Mallya and Lazebnik [2018], underline that neural networks are heavily over-parameterized. Their parameters are used inefficiently, and they can be pruned significantly without large performance drops. In lifelong learning and multi-task learning it is essential to use the network efficiently, in order to limit computational overhead when introducing new tasks. There are recent works that achieve good performance in these tasks by introducing modulation (respectively, adaptation) strategies. In particular, Li et al. [2018], Takeda et al. [2021] propose an efficient lifelong learning / domain adaptation method for multi-task learning, utilizing single feature-wise linear modulation (FiLM).

Inspired by that line of work, we propose a new, efficient ensemble method, FiLM-Ensemble. Our method adapts feature-wise linear modulation as an alternative way to construct an ensemble for (epistemic) uncertainty estimation. FiLM-Ensemble greatly reduces the computational overhead compared to the naïve ensemble approach, while performing almost on par with it, sometimes even better. In a nutshell, FiLM-Ensemble can be described as an implicit model ensemble in which each individual member is defined via its own set of linear modulation parameters for the activations, whereas all other network parameters are shared among ensemble members – and therefore only need to be stored and trained once. Thanks to this design, our method requires only a very small number of additional parameters on top of the base network, e.g., converting a single ResNet-18 model to an ensemble of 16 models increases the parameter count by 1.3%, compared to an increase by 1500% when setting up a naïve ensemble, see Table 1. We further show that FiLM-Ensemble results in more diverse ensemble members compared to other efficient ensemble methods. For instance, on the Cifar-10 benchmark it achieves diversity scores $> 6.9\%$ and up to $9.2\%$ (depending on ensemble size), against $6.8\%$ for a naive ensemble, see Fig. 1. Our contributions can be summarized as follows.

- We propose FiLM-Ensemble, a novel parameter- and time-efficient deep ensemble method.

- FiLM-Ensemble is designed in such a way that it can be readily combined with many popular deep learning models, by simply replacing batch normalisation (batchnorm) layers with conditional batchnorm layers.

- We show that FiLM-Ensemble provides an excellent trade-off between accuracy and calibration performance, improving over existing ensemble methods.

## 2 FiLM-Ensemble

Perez et al. [2018] achieved the linear feature modulation by varying the affine parameters of the batch normalization layers $\boldsymbol{\gamma}$ and $\boldsymbol{\beta}$. For each batchnorm layer $n$ in the network, the parameters are predicted according to some conditioning input $z$ (for instance, different prediction tasks):

$$\boldsymbol{\gamma}_n = g_n(z) \qquad \boldsymbol{\beta}_n = h_n(z) \tag{1}$$

The size of $\boldsymbol{\gamma}_n$ and $\boldsymbol{\beta}_n$ is $\mathbb{R}^{D_n}$ where $D_n$ is the feature dimension at layer $n$. These parameters are then used to linearly modulate the activations at the $n$-th layer $\mathbf{F}_n$:

$$\text{FiLM}(\mathbf{F}_n|\boldsymbol{\gamma}_n, \boldsymbol{\beta}_n) = \boldsymbol{\gamma}_n(z) \circ \mathbf{F}_n + \boldsymbol{\beta}_n(z) \ , \tag{2}$$

with $\circ$ being the Hadamard (element-wise) product taken w.r.t. the feature dimension.

In our scenario we aim to use the affine parameters to instantiate different ensemble members, i.e., the conditioning variable $z$ is simply an index $m \in \{1, ..., M\}$ that identifies the ensemble member. The functions $g_n(z)$ and $h_n(z)$ degenerate to look-up tables, so we can dispose of them and simply learn $M$ different sets of affine parameters for every batchnorm layer $\mathbf{F}_n$:

$$\text{FiLM}(\mathbf{F}_n|\boldsymbol{\gamma}_n^m, \boldsymbol{\beta}_n^m) = \boldsymbol{\gamma}_n^m \circ \mathbf{F}_n + \boldsymbol{\beta}_n^m \ . \tag{3}$$

To achieve ensemble members with diverse parameters, we resort to Xavier initialization, i.e., all $\boldsymbol{\gamma}_n^m$ and $\boldsymbol{\beta}_n^m$ are sampled from a uniform distribution bounded between:

$$\pm \frac{\sqrt{3}}{\sqrt{D_n}} \rho \ , \tag{4}$$

where $D_n$ is the number of features (channels) in the $n$-th layer, and $\rho$ is an initialization gain factor. In our setting $\rho$ is a tunable hyperparameter that allows one to control the trade-off between predictive accuracy and calibration of the model, see Section 3.5. In general, increasing $\rho$ leads to more diverse ensemble members and thus favours calibration. Note that as $\rho \to 0$, all ensemble members start from similar initial values for $\boldsymbol{\gamma}_n^m$ and $\boldsymbol{\beta}_n^m$ and the FiLM-Ensemble gradually collapses to a single model.

During training, we feed each input sample $\mathbf{x}$ to every ensemble member and obtain predictions $y_m = f_{\boldsymbol{\theta}, \boldsymbol{\gamma}^m, \boldsymbol{\beta}^m}(\mathbf{x})$, which depend both on the member-specific parameters $(\boldsymbol{\beta}^m, \boldsymbol{\gamma}^m)$ and on the shared parameters $\theta$. All those parameters are optimized together to minimise the chosen loss function. Here we focus on classification and use a standard cross-entropy (CE) loss.

At inference time the final prediction $\hat{y}$ is obtained by averaging the $M$ predictions of the ensemble:

$$\hat{y} = \frac{1}{M} \sum_{m=1}^{M} y_m \ . \tag{5}$$

### 2.1 Implementation Details

We implemented 1-dimensional and 2-dimensional FiLM-Ensemble layers, to be used in combination with 1-dimensional and 2-dimensional convolution operations, respectively. The forward passes through different ensemble members can easily be parallelized by replicating both the input tensor and the FiLM parameters $\boldsymbol{\gamma}, \boldsymbol{\beta}$ along the batch dimension and applying the affine transformation Eq. 3 (the same holds for the BatchEnsemble method). In this way all ensemble members run simultaneously on a single device, thus optimally utilizing modern tensor computing hardware without having to load several instances of the classification network into memory. We have implemented FiLM-Ensemble in PyTorch and release the source code.[1]

For all experiments, we optimize the model parameters with standard stochastic gradient descent, with momentum $\mu = 0.9$ and weight decay $\lambda = 0.0005$ for regularization. We train for 200 epochs with batch size 128. The learning rate is initially set to 0.1 and decays according to a cosine annealing schedule [Loshchilov and Hutter, 2017]. The initialisation gain is set to $\rho = 2$ for all experiments, except for genome sequences (see Table 3), where $\rho \in \{4, 8, 16, 32\}$. Please refer to the supplementary material for additional, dataset-specific details.

---

[1]`https://github.com/prs-eth/FILM-Ensemble`

## 3 Experiments

In the following, we first empirically examine the ability of FiLM-Ensemble to learn independent ensemble members, and compare it to deep ensembles. Then we analyze the predictive accuracy and uncertainty calibration of our method, as well as its computational cost and memory consumption. We compare its performance against several baseline methods and state-of-the-art alternatives on a variety of different datasets, and using different backbone architectures. All experimental results are averaged over three runs with different random seeds.

We evaluate our proposed method on a diverse set of classification tasks, including popular image classification benchmarks, image-based medical diagnosis, and genome sequence analysis. **CIFAR-10** and **CIFAR-100** [Krizhevsky, 2009] are widely used testbeds for image classification, and deep learning in general. They consist of clean images of objects from 10, respectively 100, different semantic classes. Each of the two datasets contains 60,000 images, of which 10,000 are reserved for testing. The semantic classes are uniformly distributed in the datasets and stratified across the train/test spit. **Retina Glaucoma Detection** [Diaz-Pinto et al., 2019] is a real-world clinical dataset that includes microscopic retina images from 956 patients with the neuropathic disease Glaucoma, and from 1401 subjects with normal (healthy) retinas. Each input sample is a single RGB image; we resize all images to $128 \times 128$. The image augmentation applies by combination of crop, horizontal flip, and color jitter. **REFUGE 2020** [Orlando et al., 2020] was a challenge at the MICCAI-2020 conference, aimed at retinal Glaucoma diagnosis. The dataset consists of 800 microscopic retina images with size $1411 \times 1411$ pixels, collected from different clinics. We use this dataset in conjunction with the models trained on the previous dataset [Diaz-Pinto et al., 2019], to evaluate the ability to detect out-of-distribution samples with the help of the predicted uncertainties (Table 4). **6mA Identification** [Li et al., 2021], is a 1-dimensional sequential genome dataset. It consists of DNA sequences of rice plants, along with binary labels that indicate whether the sequence is a N6-methyladenine (6mA) site. 6mA is an important DNA modification associated with several biological processes, such as regulating gene transcription, DNA replication and DNA repair [Campbell and Kleckner, 1990, Cheng et al., 2016, Pukkila et al., 1983]. Each sequence consists of 41 nucleotides. As there are 4 different types of nucleotides, each one-hot encoded sequence is represented by a $41 \times 4$ matrix. In total there are 269,500 training samples and 38,500 test samples.

We compare FiLM-Ensemble against *(i)* a single model without any ensembling, as the elementary baseline, *(ii)* a naïve deep ensemble [Lakshminarayanan et al., 2017b], and *(iii)* MC-Dropout [Gal and Ghahramani, 2016]. Furthermore, we compare with other state-of-the-art methods, including: *(iv)* Masksemble [Durasov et al., 2021] which can be seen as an extension of MC-Dropout, *(v)* MIMO ("multi-input multi-output") [Havasi et al., 2021] which defines a multi-head architecture where each head acts as one ensemble member, and *(vi)* BatchEnsemble [Wen et al., 2020] which creates an efficient ensemble by expanding the layer weights using low rank matrices. Arguably, this layer-wise modification of an underlying, common representation is the approach that comes closest to our work.

### 3.1 Diversity Analysis

Diversity is the key to constructing powerful ensembles: nothing is to be gained from highly correlated ensemble members that return similar outputs for (almost) any input [Zhang and Ma, 2012]. In order to analyze the diversity (respectively, the degree of independence) among members, we compute two distance metrics between the members' predictive distributions. Let $f_i$ and $f_j$ denote two different ensemble members for a classification task. We first measure the *disagreement score* $\mathcal{D}$, defined as the fraction of all $s$ test samples for which the two members return different answers, averaged over all possible pairs of members:

$$\mathcal{D} = \frac{2}{M(M-1)} \sum_{i=1}^{M} \sum_{j=i+1}^{M} \sum_{k=1}^{s} \frac{1}{s} \big[ f_i(\mathbf{x}_k) \neq f_j(\mathbf{x}_k) \big] \,, \qquad (6)$$

with $[\cdot]$ being the Iverson bracket. Second, we also measure the Kullback–Leibler (KL) divergence between the two predictive distributions $p(f_i)$ and $p(f_j)$, again averaged over all pairs:

$$KL = \frac{2}{M(M-1)} \sum_{i=1}^{M} \sum_{j=i+1}^{M} \sum_{k=1}^{s} p(f_i(\mathbf{x}_k)) \log \frac{p(f_i(\mathbf{x}_k))}{p(f_j(\mathbf{x}_k))} \,. \qquad (7)$$

Table 1: Memory and inference complexity comparison (CIFAR-10/100 datasets): Number of additional trainable parameters to have 16 ensemble members for different backbones. The inference time (mult-adds) shown corresponds to the mean GPU time (number of multiply-add operations) required to run a forward pass for a batch of size 1 with 16 ensemble members. The bottom section comprises methods whose forward and backward passes are implemented in parallel over ensemble members. Measurements are done on an NVIDIA GeForce GTX 1080 Ti.

| Method | Parameters ($\downarrow$) | | Inference time (ms) ($\downarrow$) | | Mult-adds (B) ($\downarrow$) | |
|---|---|---|---|---|---|---|
| Backbone | VGG-11 | ResNet-18 | VGG-11 | ResNet-18 | VGG-11 | ResNet-18 |
| MC-Dropout | 0.0% | 0.0% | $16\times 2.5$ | $16\times 2.4$ | $16\times 0.15$ | $16\times 0.56$ |
| Deep Ensemble | 1500% | 1500% | $16\times 2.3$ | $16\times 1.8$ | $16\times 0.15$ | $16\times 0.56$ |
| Masksemble | 0.0% | 0.0% | 20.8 | 6.6 | 2.45 | 8.89 |
| MIMO | 1.1% | 0.9% | 2.7 | 1.9 | 0.18 | 0.58 |
| BatchEnsemble | 1.4% | 5.2% | 2.8 | 5.2 | 2.44 | 8.89 |
| FiLM-Ensemble | 0.9% | 1.3% | 2.8 | 5.7 | 2.45 | 8.89 |

For both metrics, higher numbers correspond to higher diversity. See Fig. 1. We observe that the average diversity increases with the number of ensemble members. In both metrics, FiLM-Ensemble achieves higher scores than the naïve, explicit deep ensemble. Meaning that the predictions of FiLM-Ensemble are less correlated than those of an equivalent number of networks trained independently with different random seeds. Also, note that with the increasing number of members, improvements in diversity metrics for the naïve explicit deep ensemble are negligible.

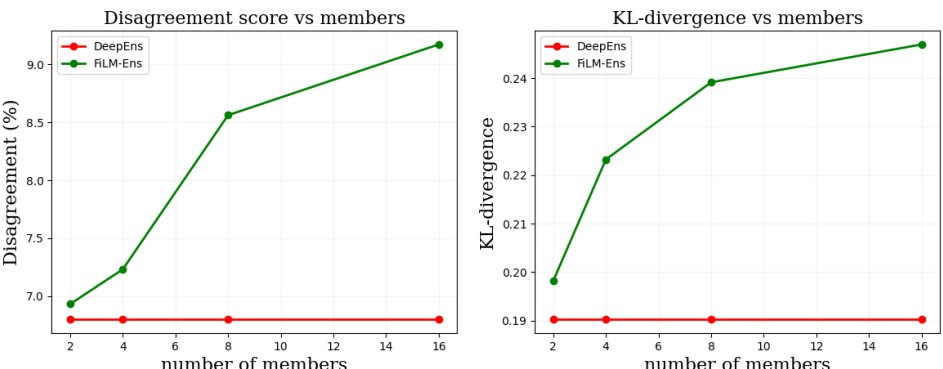

Figure 1: Diversity analysis: CIFAR-10/VGG-11 experiment. With increasing number of members, FiLM-Ensemble achieves more diverse representations. See Section 3.1.

## 3.2 Computational Cost

We go on to measure the efficiency of our proposed method, both in terms of parameter count and in terms of resources needed for a forward pass. Table 1 (left section) reports the numbers of additional numbers of parameters compared to a single network (i.e., without uncertainty calibration). Due to their design, MC-Dropout and Masksemble do not require any additional parameters. Among the remaining methods, FiLM-Ensemble has the lowest (VGG-11) or second-lowest (Resnet-18) overhead in terms of parameter count, while performing significantly better, as we will see below. In terms of inference complexity (Tab. 1, center and right), FiLM-Ensemble turns out to be very competitive. Only MIMO with a ResNet-18 backbone is significantly faster and lighter, however this comes at a considerable price in terms of accuracy and calibration, see below.

## 3.3 CIFAR-10 / CIFAR-100

We perform several experiments on widely used benchmarks in computer vision, CIFAR-10 and CIFAR-100. As performance metrics, we plot the test set accuracy and the expected calibration error

Table 2: Classification performance on CIFAR-100 with $M = 4$, using ResNet-18/34 as backbone. Best score for each metric in **bold**, second-best underlined.

| Backbone | Resnet-18 | | Resnet-34 | |
|---|---|---|---|---|
| Method | Acc ($\uparrow$) | ECE ($\downarrow$) | Acc ($\uparrow$) | ECE ($\downarrow$) |
| Single Network | 78.0 $\pm$0.4 | 0.046 $\pm$0.001 | 79.3 $\pm$0.2 | 0.089 $\pm$0.006 |
| Deep Ensemble | **81.6** $\pm$0.3 | 0.041 $\pm$0.002 | **82.0** $\pm$0.1 | **0.044** $\pm$0.002 |
| MC-Dropout | 75.5 $\pm$0.6 | 0.064 $\pm$0.003 | 72.2 $\pm$0.2 | 0.079 $\pm$0.004 |
| MIMO | 48.0 $\pm$2.6 | 0.083 $\pm$0.017 | 56.2 $\pm$4.8 | 0.132 $\pm$0.055 |
| Masksemble | 72.5 $\pm$0.5 | 0.075 $\pm$0.004 | 70.1 $\pm$1.2 | 0.067 $\pm$0.004 |
| BatchEnsemble | 77.7 $\pm$ 0.1 | 0.052 $\pm$ 0.002 | 78.3 $\pm$ 0.1 | 0.056 $\pm$ 0.002 |
| FiLM-Ensemble | 79.4 $\pm$0.2 | **0.038** $\pm$0.000 | 80.2 $\pm$0.1 | 0.045 $\pm$0.001 |

(ECE) against the ensemble size $M$ in Fig. 2, for all compared methods. In terms of accuracy (left subfigure), FiLM-Ensemble is outperformed only by the explicit deep ensemble, across all tested values of $M \in \{2, 4, 8, 16\}$. These two surpass all other methods by a clear margin. Surprisingly, we observe that BatchEnsemble generally exhibits a negative correlation between ensemble size and test set accuracy, with $M = 4$ performing best among all settings. MIMO shows very poor performance in this experimental setting in terms of accuracy (and also in ECE). We speculate that its shared backbone probably has a tendency to fragment into largely independent ensemble members of low channel depth, which lack the necessary capacity when using comparatively small networks like VGG-11 or Resnet-18. With regard to ECE (right subfigure), we see all methods improving with growing ensemble size $M$. FiLM-Ensemble achieves better calibration then the widely used deep ensemble and MC-dropout methods, with significant margins at large ensemble sizes. Masksemble and BatchEnsemble achieve very good calibration for sizes $M \in [4 \ldots 16]$, but at the cost of significant drops in classification performance. An attractive feature of FiLM-Ensemble is that it offers a simple mechanism for trading off accuracy against calibration, by tuning the gain factor $\rho$. See Section 3.5.

In Tab. 2 we quantitatively compare all tested methods on the CIFAR-100 dataset, both with a standard Resnet-18 backbone and with a larger Resnet-34. We observe that with both architectures, and in terms of both predictive accuracy and calibration, FiLM-Ensemble always either ranks first, or it ranks second behind the inefficient, explicit deep ensemble. Also, note that when the capacity of the network is increased (Resnet-18 $\rightarrow$ Resnet-34), calibration performance improves for some of the implicit methods, but at the cost of reduced accuracy. Overall, the experiments with both variants of CIFAR and all three model architectures confirm that our method presents an excellent trade-off between predictive accuracy, uncertainty calibration and computational efficiency.

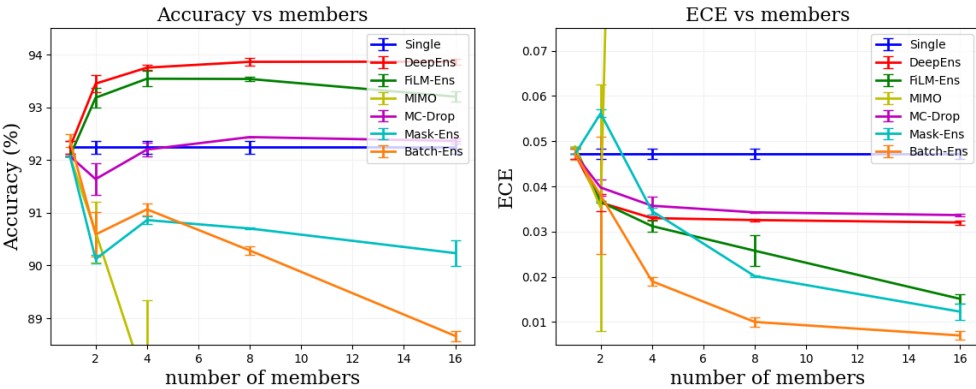

Figure 2: Accuracy and ECE for CIFAR-10, with varying ensemble sizes, using VGG-11 as backbone.

Table 3: Classification performance for retinal Glaucoma images. Best score for each metric in **bold**, second-best underlined.

| Method | Accuracy (%) (↑) | | | | ECE (↓) | | | |
|---|---|---|---|---|---|---|---|---|
| # member ($M$) | 2 | 4 | 8 | 16 | 2 | 4 | 8 | 16 |
| Single | 84.4 | | | | 0.084 | | | |
| Deep Ensemble | $85.6 \pm 0.2$ | $85.7 \pm 0.3$ | $86.0 \pm 0.2$ | $86.8 \pm 0.4$ | $0.041 \pm 0.002$ | $0.078 \pm 0.002$ | $0.091 \pm 0.004$ | $0.066 \pm 0.003$ |
| FSSD Huang et al. [2020] | $85.9 \pm 0.1$ | | | | $0.047 \pm 0.002$ | | | |
| SNGP Liu et al. [2020] | $84.7 \pm 0.2$ | | | | $0.064 \pm 0.003$ | | | |
| pNML Bibas et al. [2021] | $85.6 \pm 0.1$ | | | | $0.061 \pm 0.001$ | | | |
| MC-Dropout | $67.0 \pm 0.2$ | $78.4 \pm 0.5$ | $80.0 \pm 0.4$ | $82.7 \pm 0.4$ | $\mathbf{0.002} \pm 0.001$ | $0.046 \pm 0.009$ | $0.053 \pm 0.011$ | $0.051 \pm 0.018$ |
| MIMO | $72.4 \pm 1.9$ | $69.8 \pm 2.3$ | $68.9 \pm 2.4$ | $68.3 \pm 2.4$ | $0.049 \pm 0.011$ | $0.061 \pm 0.013$ | $0.082 \pm 0.017$ | $0.041 \pm 0.019$ |
| Masksemble | $82.7 \pm 0.5$ | $83.0 \pm 0.5$ | $80.2 \pm 0.6$ | $81.7 \pm 1.1$ | $0.064 \pm 0.004$ | $0.049 \pm 0.007$ | $\underline{0.021} \pm 0.010$ | $0.062 \pm 0.012$ |
| BatchEnsemble | $84.5 \pm 0.1$ | $86.5 \pm 0.1$ | $86.8 \pm 0.2$ | $\underline{87.1} \pm 0.2$ | $0.035 \pm 0.003$ | $0.063 \pm 0.009$ | $0.071 \pm 0.002$ | $0.066 \pm 0.002$ |
| FiLM-Ensemble | $86.3 \pm 0.1$ | $86.8 \pm 0.2$ | $86.9 \pm 0.1$ | $\mathbf{87.8} \pm 0.1$ | $0.062 \pm 0.001$ | $0.074 \pm 0.000$ | $0.068 \pm 0.002$ | $0.055 \pm 0.001$ |

## 3.4 Retinal Glaucoma Detection

Glaucoma is currently the leading reason of irreversible blindness in the world. Detection of glaucomatous structural damages and changes is a challenging task in the field of ophthalmology. We evaluate our proposed FiLM-Ensemble, as well as the baselines described above, on the ask of diagnosing Glaucoma and quantifying the uncertainty associated with the prediction, see Table 3. The proposed FiLM-Ensemble achieves the best classification result across all ensemble sizes, and also the best overall result, with $M = 16$. Whereas there is no clear trend with respect to uncertainty calibration.

## 3.5 6mA Identification

With the 6mA identification task we show that FiLM-Ensemble can also be readily combined with existing models for 1-dimensional sequential genome data. We use the 6mA-rice-Lv dataset and a 1D-CNN architecture whose hyper-parameters have already been optimized for this dataset [Li et al., 2021]. FiLM-Ensemble improves the accuracy and the calibration of that model, see Fig.3. Our method performs on par with the explicit deep ensemble and better than a single instance of the model tuned for the specific task. More importantly, one can reach a significantly better calibration (lower ECE) by increasing the gain $\rho$, with only a minimal accuracy drop by $< 0.25$ percent points. Please refer to the Section **??** for more results.

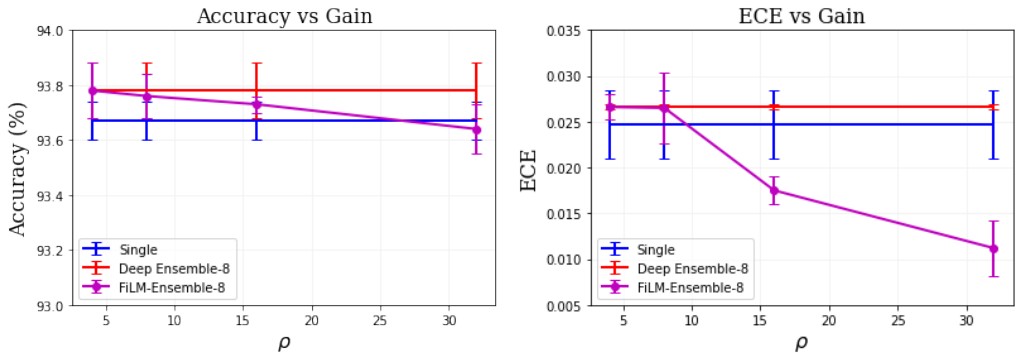

Figure 3: Performance of FiLM-Ensemble with varying gain $\rho$, c.f. Section 3.5.

## 3.6 Out-of-Distribution Detection

Domain shift often occurs in medical datasets; thus, detecting out-of-distribution (OOD) samples is an important task in clinical diagnosis. Model uncertainty can be used for OOD detection. For our experiment, we regard the retinal Glaucoma dataset [Diaz-Pinto et al., 2019] as the within-distribution samples and perform OOD detection with the REFUGE dataset [Orlando et al., 2020]. As performance metric, we report AUROC (Area Under the Receiver Operating Characteristic) scores, Table 4. FiLM-Ensemble can detect OOD test samples with significantly higher accuracy than other

Table 4: OOD detection for retinal Glaucoma images. Best score in **bold**, second-best underlined.

| Method | AUROC (%) (↑) | | | |
|---|---|---|---|---|
| # member | 2 | 4 | 8 | 16 |
| Single | 68.42 | | | |
| Deep Ensemble | $76.89 \pm 0.1$ | $77.91 \pm 0.2$ | $78.22 \pm 0.1$ | $78.06 \pm 0.1$ |
| FSSD Huang et al. [2020] | $76.42 \pm 0.2$ | | | |
| SNGP Liu et al. [2020] | $71.25 \pm 0.6$ | | | |
| pNML Bibas et al. [2021] | $76.68 \pm 0.3$ | | | |
| MC-Dropout | $68.03 \pm 0.3$ | $69.79 \pm 0.2$ | $77.94 \pm 0.6$ | $72.22 \pm 0.2$ |
| MIMO | $57.33 \pm 1.4$ | $59.49 \pm 1.2$ | $61.74 \pm 2.1$ | $60.52 \pm 3.1$ |
| Masksemble | $71.22 \pm 0.5$ | $70.83 \pm 0.8$ | $72.04 \pm 1.1$ | $70.95 \pm 1.4$ |
| BatchEnsemble | $74.38 \pm 0.1$ | $72.61 \pm 0.3$ | $75.44 \pm 0.2$ | $75.04 \pm 1$ |
| FiLM-Ensemble | $77.02 \pm 0.1$ | $77.92 \pm 0.2$ | $\underline{79.43} \pm 0.1$ | $\mathbf{79.85} \pm 0.2$ |

efficient ensemble methods, standard state-of-the-art OOD detection methods, and even the explicit deep ensemble. This suggests that for challenging test samples, which are not adequately represented in the training data, the uncertainties estimated with FiLM-Ensemble are better calibrated.

# 4 Related Work

## 4.1 Epistemic Uncertainty Quantification

A large corpus of related work addresses the estimation of epistemic (model) uncertainty in neural networks. At the heart of such modeling is often the concept of *marginalization instead of optimization*, i.e., integrating out a (possibly uncountably infinite) set of models weighted by their posterior probability, instead of committing to a point estimate of that distribution. A multitude of methods have been proposed to implement approximate Bayesian inference w.r.t. the model weights, given the training data and an appropriate prior, a process that is not analytically tractable in general [Kendall and Gal, 2017].

For instance, methods based on *variational inference* [Graves, 2011, Ranganath et al., 2014, Blundell et al., 2015] seek to learn an approximate posterior distribution which is a member of a simpler family of variational distributions. This variational distribution can often be learned using backpropagation [Blundell et al., 2015] and can then be sampled from, or sometimes even used for exact inference. Markov chain Monte Carlo sampling (MCMC) approaches [Neal, 1996, Welling and Teh, 2011, Chen et al., 2014] construct a Markov chain which has the *exact* posterior distribution as its stationary distribution, that can then be employed for sampling. However, in practise, these approaches often fail to sufficiently explore high-dimensional, multi-modal loss landscapes as they are common in deep learning [Gustafsson et al., 2020].

## 4.2 Ensembles and Sub-networks

In a method referred to as *deep ensembles* [Lakshminarayanan et al., 2017a], a set of $M$ neural network models are randomly and independently initialized, and are subjected to stochastic mini-batch sampling during SGD training. The models generally converge to different minima in the parameter space, and can be considered samples from an approximate posterior [Wilson and Izmailov, 2020, Gustafsson et al., 2020, Izmailov et al., 2021]. Deep ensembles often achieve the best calibration and predictive accuracy [Ovadia et al., 2019, Gustafsson et al., 2020, Ashukha et al., 2020], but suffer from high computational complexity as they require training, storing, and running inference on several full instances of the network.

Many alternative methods have recently been proposed in an attempt to reduce either the computational cost or the storage cost of deep ensembles. Monte Carlo (MC) Dropout [Gal and Ghahramani, 2016] runs multiple forward passes on the dropout layers in order to obtain multiple predictions and obtain an uncertainty estimate. Although the method requires fewer computations compared to deep ensembles, it also leads to less accurate uncertainty estimates. Snapshot ensembles [Huang

et al., 2017] use a cyclic learning rate schedule in order to find multiple local minima, they then store multiple copies of the network to create the ensemble. While this approach reduces the computation cost during training it does not alleviate the storage cost.

BatchEnsembles [Wen et al., 2020] employ multiple low rank matrices, which can be stored efficiently, in order to modulate the parameters of a neural network and thus mimic an ensemble of network models. Masksembles [Durasov et al., 2021] aim to improve the performance of MC Dropout by carefully selecting the dropout masks, used to drop certain features, such that they lead to better uncertainty quantification. Another approach that uses multiple sub-networks is proposed in Havasi et al. [2021]. In this case additional, independent layers are added at the beginning and at the end of the network, in order to obtain multiple prediction with an single backbone model. Although such method seems to reduce the computational resources required at training and inference time, the gain is limited to larger backbones, such as Wide-ResNet (e.g., ResNet28-10), whereas for widely used standard architectures like VGG (e.g., -16), or ResNet (e.g., -34), the approach is not very effective, as shown in our experiments.

Although many methods have been proposed that aim at reducing the computational cost of deep ensembles, none of them appears to clearly outperform most others. In summary, the question how to effectively model uncertainty still remains open.

### 4.3 Feature-wise Linear Modulation

The idea of controlling the batch normalization parameters to modulate a network's function has been explored by many different authors to accomplish different tasks. Conditional batch normalization (CBN) was proposed by de Vries *et al.*, who achieved good performance in VQA experiments by using an MLP to estimate residuals for normalization parameters used in a pre-trained ResNet [De Vries et al., 2017b]. Perez et al. [2018] proposed FiLM also for solving VQA tasks. Strub et al. [2018] modify FiLM to produce normalization parameters in several stages instead of all at once. This formulation is better able to handle longer conditioning information, such as dialogues instead of questions, and achieved excellent results on the GuessWhat?! visual dialogue task [De Vries et al., 2017a].

Such modulation operations have also been used for image style transfer. It has been observed that the statistics of a feature map, which are associated with and can be controlled by the parameters produced by FiLM, are directly related to the style of an image [Huang and Belongie, 2017]. Dumoulin et al. [2017] succeeded in capturing the styles of artistic paintings, as well as combining extracted styles to create new ones, by using conditional instance normalization, which can be seen as a variation of FiLM. Ghiasi et al. [2017] expand on this work by jointly training a style prediction network and a style transfer network, which also operate based on conditional instance normalization. Finally, Brock et al. [2018] used FiLM for natural image synthesis using generative adversarial networks (GANs). They report that this allowed for a reduction in computation and memory costs, as well as a 37% increase in training speed.

Yang et al. [2018] have used FiLM to modulate the layers of a segmentation network to perform video object segmentation. This made it possible to avoid the fine-tuning process that was used by competing methods, which resulted in a $70\times$ speed-up while achieving similar accuracy. Feature modulation has also been applied for various other tasks. Oreshkin et al. [2018] used FiLM for task-dependent metric scaling, which allowed them to achieve excellent results in few-shot classification. Vuorio et al. [2019] use FiLM for meta-learning via task-aware modulation. The authors note that FiLM outperforms attention-based modulation in this context, and is more stable. Finally, Vinyals et al. [2019] used FiLM in their AlphaStar neural architecture for multi-agent reinforcement learning. To our knowledge, FiLM has so far not been used for (implicit) model ensembling or uncertainty quantification.

## 5 Limitations & Future Work

The work presented in this paper gives rise to several questions that can be explored in future work. Using pre-trained models is standard procedure in deep learning applications. It would be useful to explore how FiLM-Ensemble performs when used in conjunction with pre-trained models. This may not be straightforward since FiLM-Ensemble (and also other implicit ensemble methods,

e.g., BatchEnsemble, MIMO) is based on variations between ensemble members, which may be reduced if all members are similarly initialized. Furthermore, self-attention-based models, e.g., Transformers [Vaswani et al., 2017] and Vision Transformers [ViTs, Dosovitskiy et al., 2021] have recently become very popular; therefore it is natural to adapt FilM-Ensemble to work with such models. Note that layer normalization is standard in Transformers instead of batch normalization, which prevents the straightforward application of the presented method in this case. Also, a number of measures designed to enhance implicit ensembles are orthogonal to our approach and could potentially be combined with FiLM-Ensemble to further improve its performance and uncertainty calibration, while minimally increasing the computational costs. It appears straight-forward to add the selection of independent examples for each member during training [as in MIMO, Havasi et al., 2021], temperature scaling [Guo et al., 2017] or data augmentation strategies [such as, e.g., Ramé et al., 2021].

## 6 Conclusion

In this paper we present FiLM-Ensemble, a novel implicit deep ensembling method. We achieve high efficiency using a simple yet effective idea – feature-wise linear modulation – which has been shown to be effective in different domains, such as image style transfer, model-agnostic meta-learning, or multi-task learning. Our extensive evaluation shows that FiLM-Ensemble outperforms, or is on par with, state-of-the-art ensemble methods in many different experimental settings.

## Broader Impact

Machine learning has recently witnessed a steep increase in model sizes and associated computational costs, and as a consequence a rapid growth in energy consumption. For instance, training state-of-the-art language models like GPT-3 would amount to at least 1400 MWh, or 4.6 million $. Therefore, recently the term *Green AI* has been introduced, referring to AI research that yields novel results while taking into account the computational cost, encouraging a reduction in resources spent. To this extent, we believe that the presented method can foster more efficient ensemble methods and be helpful towards a greener AI.

**Acknowledgements** We would like to thank Liyuan Zhu for his invaluable contributions to experiments. H.A.G. received funding from German Federal Ministry of Education and Research (BMBF, Grant "GenomeNet" 031L0199B). B. B. and M. R. were supported by the Bavarian Ministry of Economic Affairs, Regional Development and Energy through the Center for Analytics - Data - Applications (ADACenter) within the framework of BAYERN DIGITAL II (20-3410-2-9-8) and the German Federal Ministry of Education and Research and the Bavarian State Ministry for Science and the Arts.

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
