# A  Training Details

## A.1  CIFAR-10 / CIFAR-100

Please refer to Section 2.1.

## A.2  Retinal Glaucoma Detection

We followed the same training procedure as for CIFAR-10/100, please refer to Section 2.1. Resnet-18 is used as a backbone. Models are trained with 70% of the dataset for 150 epochs and tested on the test set (30% of the dataset).

## A.3  6mA Identification

A 1-dimensional CNN architcture is used, whose hyperparameters such as kernel size and the number of layers are optimized by Li et al. [2021]. The CNN consists of 5 convolutional blocks, where each block contains a 1-dimensional convolution, ReLU activation, batch normalization, and dropout with a rate of 0.5. The convolutional layers have a filter size of 256, kernel size of 10, a stride of 1, and the first convolutional layer have a padding of 5. On top of the last convolutional block, there is a linear layer for predicting the binary labels. Binary cross-entropy is used as a loss. All models are trained for 20 epochs. The initial learning rate of 0.01 is used, as in Li et al. [2021]. Cosine-annealing is used as a learning rate scheduler.

# B  Additional Results

## B.1  EfficientNet as Backbone & More Calibration Metrics

We run more experiments using more modern architecture: EfficientNet Tan and Le [2019] whose proportion of the number of channels vs. the number of layers can vary drastically, compared to Resnets. In this experiment we use two extra calibration metrics: (i) the Brier score Brier et al. [1950] and (ii) the Static Calibration Error (SCE) Nixon et al. [2019]. SCE can be considered an extension of ECE but more accurately account for calibration by considering all classes, instead of just the one with the highest confidence. Table 1 shows that FiLM-Ensemble can also be effectively used in conjunction with EfficientNet architecture. In addition, other calibration metrics are also in favor of FilM-Ensemble.

Table 1: CIFAR-10/EfficientNet-B0 performance comparison. $M \in \{2, 4\}$. The best score for each metric is printed **bold**.

| Method | Acc ($\uparrow$) | ECE ($\downarrow$) | SCE ($\downarrow$) | Brier ($\downarrow$) |
|---|---|---|---|---|
| Single | 90.80 | 0.0496 | 0.0106 | 0.1470 |
| MC-Dropout (2) | 90.81 | 0.0499 | 0.0107 | 0.1478 |
| MC-Dropout (4) | 90.81 | 0.0497 | 0.0107 | 0.1474 |
| Deep Ensemble (2) | 92.67 | 0.0373 | 0.0080 | 0.1146 |
| Deep Ensemble (4) | **93.30** | 0.0307 | 0.0067 | **0.1008** |
| Film-Ensemble (2) | 91.62 | 0.0336 | 0.0073 | 0.1291 |
| Film-Ensemble (4) | 91.73 | **0.0163** | **0.0044** | 0.1222 |

## B.2  Calibration-Accuracy Trade-off

As in Section 3.5, we show that one can reach a significantly better calibration (lower ECE) by increasing the gain $\rho$, with only a minimal accuracy drop. In this case, we use Resnet-34 with 2 ensemble members on Cifar-100 dataset. See Fig. 1. Also note Fig. 2 is an extension of Fig. 3 (of the main text) with various number of ensemble members.

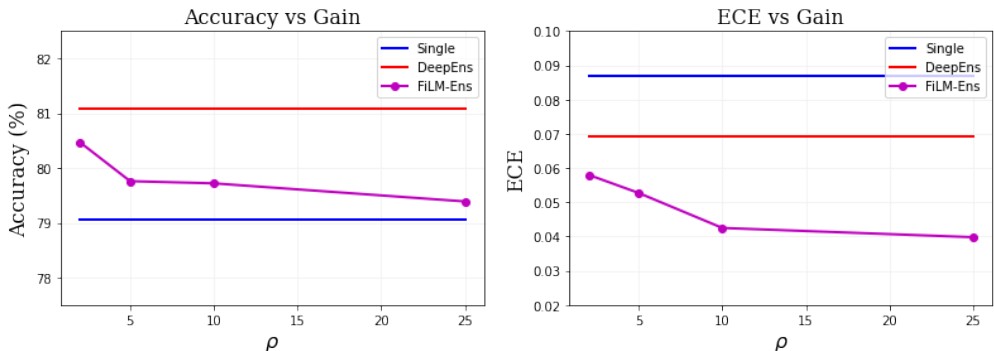

Figure 1: Performance of FiLM-Ensemble with varying gain $\rho$ on Cifar-100 using Resnet-34 as backbone with $M = 2$, c.f. Section B.2.

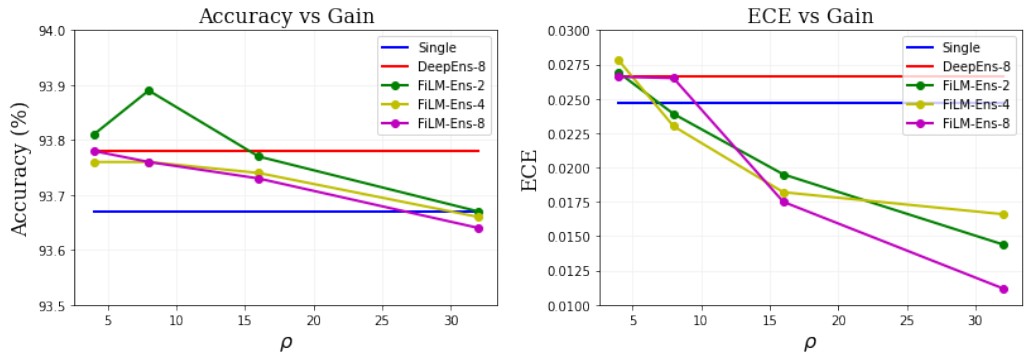

Figure 2: Performance of FiLM-Ensemble with varying gain $\rho$ on 6mA-rice-Lv dataset, using CNN-based Deep6mA as backbone with $M \in \{2, 4, 8\}$, c.f. Section 3.5.