# OpenReview forum: "FiLM-Ensemble: Probabilistic Deep Learning via Feature-wise Linear Modulation"
_NeurIPS.cc/2022/Conference — NeurIPS 2022 Accept_

### Official Review · Reviewer_4hVr · 2022-07-01

**Rating:** 6
**Confidence:** 5
**Soundness:** 3 good
**Presentation:** 3 good
**Contribution:** 3 good

**Summary:**

The paper proposes a parameter efficient approach to ensembles based on FiLM layers. The idea is that each of the ensemble members share all of the parameters in the network with the exception of a set of the FiLM layer parameters that are specific to each member. The FiLM parameters in each ensemble member are initialized with different random values at the start of training and evolve independently over the training process. At test time, the logits of each of the ensemble members are averaged to make a prediction. The method is parameter efficient since the FiLM parameters make up only a tiny fraction of the overall network parameter count. Despite only a small number of parameters differing in each ensemble member, the authors show that the diversity of the ensemble members is high, and overall accuracy and calibration is competitive with methods that are more expensive in terms of memory and computation time.

**Questions:**

The method is quite simple and the paper was clear. As a result, there are no questions.

**Limitations:**

Despite the response in the checklist, I could not find where the authors described limitations of the proposed method. Some identified limitations of the work are described above in the **Weaknesses** section (i.e. the method was tested on relatively small networks and datasets).

The authors did not discuss any negative societal aspects of the work, which is unfortunate as image classification technology has a dark side (i.e. surveillance, military, etc.).

**Strengths And Weaknesses:**

**Strengths**
1. The paper is clear and well written.

2. The concept of using FiLM layers as the basis for an ensemble method is fundamentally good. The method is simple and effective.

3. Results overall are excellent and compare favorably to leading competitive methods.

**Weaknesses**
1. The paper evaluates on relatively small and outdated networks (VGG, ResNet18/34). The paper would be improved if larger and more up to date networks (i.e. EfficientNets, Vision Transformers, etc.) were used in the evaluation.

2. The datasets used to evaluate the method are relatively small. The paper would be improved if larger and potentially more challenging datasets/benchmarks were used (e.g. ImageNet, VTAB)

3. There are no error bars in the tables, despite the fact that each experiment was repeated 3 times with different random seeds.

4. While the out of distribution experiments compare to other ensemble methods, the paper would be improved if it compared to state-of-the-art (or even standard) OOD methods.

5. The only calibration metric used was ECE. The paper would be improved if additional metrics were calculated (Brier sore, etc.).

There are some typos:
- Line 56: na ive
- Line 107: bot
- Line 116: thid
- Line 119: and **will** release the source code

---

> ### Author Response · Authors · 2022-08-02
> **Response to Reviewer 4hVr**
>
> Thank you for the encouraging words and the detailed review! We answered your remarks point-by-point below.
>
> 1-Use larger and more up-to-date networks (i.e. EfficientNets, Vision Transformers, etc.)
> -----
> This was also asked by reviewer #xEc2, in particular w.r.t. EfficientNet’s different proportion between the number of channels and number of layers. We have implemented FilM-Ensemble, MC-Dropout, and Deep-Ensemble with EfficientNet-B0 and run experiments on the Cifar10 dataset. The results are below.
>
> |   | Acc  | ECE  | SCE  | Brier  |
> |---|---|---|---|---|
> | Single  | 90.80  |0.0496   |  0.0106	 | 0.1470  |
> | MC-Drop (2)  | 90.81   |0.0499   | 0.0107  |  0.1478 |
> | MC-Drop (4)  |90.81   | 0.0497  | 0.0107  | 0.1474  |
> | Deep Ensemble (2)  | 92.67  |  0.0373 |  0.0080 | 0.1146  |
> | Deep Ensemble (4)  | **93.30**  |  0.0307 | 0.0067  | **0.1008**  |
> | Film-Ensemble (2)  | 91.62  | 0.0336  |  0.0073 |  0.1291 |
> | Film-Ensemble (4)  | 91.73  | **0.0163**  | **0.0044**  | 0.1222  |
>
>
> Our experiments show that the proposed method also shows good performance with Efficient-Net architecture. FilM-Ensemble achieves a significantly better calibration than deep ensemble in terms of ECE and SCE while deep ensemble accuracy is a bit higher than ours. We will add these results accordingly either in the main text or supplementary for the camera-ready version.  However, it did not seem feasible to us to upscale our method to very high-parameter count transformer models, in particular in such a short time frame and given limited computational resources. We would therefore like to leave that extension to future work.
>
> 2-The datasets used to evaluate the method are relatively small
> -----
> We agree that the datasets used in our evaluation are not the largest, however we think that the datasets used are standard ones for studying uncertainty quantification. The scalability of the method in terms of dataset size is proven with Cifar10 and Cifar100 datasets. Although we would like to add an ImageNet comparison for the final camera-ready version if the time allows.
>
> 3-Missing error bars in some experiments
> -----
> Thanks for spotting this! We have added the missing error bars to the respective figures in the manuscript.
>
> 4-Comparison with dedicated OOD methods
> -----
> We performed additional experiments and compared our results for the task of OOD detection with recent state-of-the-art methods for this task. Specifically, we compared our result with pNML which is a single layer predictive normalized maximum likelihood for OOD detection and for the fair comparison, we used the same pre-trained model as single model. However, based on reported results shown in Table 3 and Table 4, FilM-Ensemble outperformed pNML. We also compared our results with the popular SNGP model which is distance-aware uncertainty with spectral-normalized neural GP (Gaussian Process) and our FilM-Ensemble achieved 8% better results when it has 16 members.  Moreover, we compare our result with FSSD and achieved 2% better accuracy and 3% better AUROC. We added the new results to the paper.
>
> 5-Additional calibration metrics
> -----
> We agree with your remark that our experimental evaluation would benefit from additional model calibration metrics. We have therefore computed both the Brier score and the Static Calibration Error (that takes into account all classes instead of just the highest-confidence class). With these extra two metrics, we have made some comparisons in multiple experimental settings and these metrics are also in favor of FilM-Ensemble. See the previous table and the tables below. We will include these metrics (including missing numbers for other baselines) in the main manuscript for the camera-ready version.
>
> Brier score for Genome dataset:
>
> |   | Brier |
> |---|---|
> |  Single |  0.0519  |
> | Deep-Ensemble (8)   | 0.0512  |
> |FiLM-Ensemble (8)   | **0.0505**  |
>
>
> Resnet34/Cifar100:
>
> |   | SCE  | Brier |
> |---|---|---|
> |  Batch-Ensemble (4) | 0.0018  | 0.3123  |
> | FiLM-Ensemble (4)  | **0.0016**  | **0.2914**  |
>
>
> 6-Limitations
> -----
> Thank you for the suggestion. Due to the page limit, we can not add more text to the manuscript at the moment but we would like to add an extra section to the final version that discusses the mentioned limitations of the proposed method accordingly.
>
> 7-Societal impacts of the work
> -----
> We take the impact of our work very seriously and are already discussing (Section 6) the field of resource-aware green AI, to which we believe our method contributes. However, we argue that the possible negative societal impacts mentioned by the reviewer are too broad and not directly related to the main topic of the paper; but we are happy to discuss that further during the discussion period.

---

> > ### Comment · Reviewer_4hVr · 2022-08-08
> > **Thanks for the response**
> >
> > Thanks for the response. The only remaining concern that I have is whether the method would be as effective when:
> >
> > 1. Using larger networks (e.g. ResNet152x4, EfficientNet-B7, EfficientNetV2-XL, ViT-Huge, etc.); and
> > 2. Larger datasets (e.g. ImageNet); and
> > 3. Greater variety of datasets (e.g. VTAB).
> >
> > Hence my score will remain at 6.

---

> > > ### Author Response · Authors · 2022-08-09
> > > **response**
> > >
> > > Thank you for your constructive comments and please find the individual answers to your raised points below:
> > >
> > > 1-Using larger networks (e.g. ResNet152x4, EfficientNet-B7, EfficientNetV2-XL, ViT-Huge, etc.)
> > > -----
> > > As suggested we implemented FiLM for EfficientNet during the rebuttal period and we presented our results for EfficientNet-B0 (table above). We would like to run more experiments for a larger version of EfficientNet and include these results in the supplementary material. In addition, we would like to draw attention to the fact that we have already shown results in a larger variety of settings (different combinations of 5 different datasets={Cifar10, Cifar100, Glaucoma, REFUGE, 6mA}, 5 different architectures={VGG11, Resnet18, Resnet34, EfficientNet-B0, Deep6mA}, and 4 different numbers of ensembles={2,4,8,16}) which is already much more than compared methods, e.g MIMO only shows results on 3 different datasets, 2 architectures and 2 different numbers of ensembles={2,3}.
> > > Regarding ViT, it is not straightforward to adapt the proposed method or other related methods, e.g Batch-Ensemble, to ViTs since they are architecture-wise quite different than CNN's - there is even no batch normalization layer used in ViTs instead layer normalization is used. This does not mean that our approach can not be adapted to ViTs, we have been working on this already (it is ongoing research itself) but we think that this should be left to future work since there is not even space to discuss what architectural tricks we made for ViTs. But here a some preliminary results for ViTs.
> > >
> > >
> > >
> > >
> > >
> > > ViT/Cifar10:
> > >
> > > | |Acc |ECE
> > > |---|---|---|
> > > Single |90.54 |0.0286
> > > FiLM-Ensemble (2) |**91.18**|**0.0269**
> > >
> > >
> > >
> > > 2-Larger datasets (e.g. ImageNet)
> > > -----
> > > Regarding larger datasets, we would like to repeat that we would like to add an ImageNet comparison for the final camera-ready version if time allows.
> > >
> > >
> > >
> > >
> > > 3- Greater variety of datasets (e.g. VTAB).
> > > -----
> > > We agree that including more datasets strengthens the paper; however; we already show results in a variety of datasets/tasks: 2 classical image classification datasets, 2 medical image datasets (1 classification and 1 OOD detection task), and 1 1-dimensional sequential genome classification dataset which is substantially more diverse than compared methods e.g MIMO use only 3 classical image classification datasets.

---

### Official Review · Reviewer_xEc2 · 2022-07-04

**Rating:** 6
**Confidence:** 4
**Soundness:** 2 fair
**Presentation:** 3 good
**Contribution:** 3 good

**Summary:**

The paper proposes the use of a set of FiLM layers instead of distinct neural networks as a way to build an ensemble. This has the benefit of drastically reducing the number of weights used by the ensemble, since it only relies on cheap channel-wise scale and shift vectors. The method has been compared with a variety of baselines (e.g. Deep Ensembles, MIMO, MC-Dropout, etc), using a few backbones (ResNets and VGG), on image-classification datasets such as CIFAR10, CIFAR100, and Glaucoma Detection. The empirical results show performances that are comparable with leading methods in terms of classification accuracy and calibration error with the benefit of a lower parameter count.

**Questions:**

1) I would like to see a discussion of the concerns that I have pointed out in the "weaknesses" section.

2) The parameters of the FiLM layers are sampled from the uniform distribution bounded by the values reported in Equation 4. I was wondering if a similar procedure has been used to produce the members of the Deep Ensemble. This is crucial, as differences in the initialization strategy could affect the performance of the baselines.

3) Assigning a different set of initial weights to the member of the ensembles is just one way to enforce diversity. Did the authors try other techniques?

**Limitations:**

The authors have adequately addressed the limitations of their work.

**Strengths And Weaknesses:**

Strengths
---------

- The method is novel and original, the use of FiLM layers to create an ensemble provides substantial memory savings.

- The method is compared against strong baselines (e.g. Deep Ensembles).

- The paper is well written, easy to follow, and clear.


Weaknesses
----------

- The empirical evaluation has been performed on relatively small networks (e.g. ResNet18 and ResNet34) and simple datasets (e.g. CIFAR10 and CIFAR100). It would be useful to verify if the same performance gains can be obtained in larger models and datasets.

- Limited variety of networks, the method has been mainly tested on ResNets. The number of parameters in a FiLM layer is related to the number of channels in that layer; in recent architectures (e.g. MobileNet and EfficientNet) the proportion of number of channels vs. number of layers can vary drastically w.r.t. ResNets. This seems quite relevant, since the total number of FiLM parameters used in the model can affect the performance, and change the gap w.r.t. the baselines (in particular Deep Ensembles). It would be useful to verify if the method works well also on other architectures.

- The use of the Expected Calibration Error (ECE). A recent line of work exposed several problems of the ECE score [1,2,3,4]. ECE is a biased estimate of the true calibration, and it only estimates miscalibration in terms of the maximum assigned probability, disregarding the full predicted probability. Other metrics have been proposed to fix those issues, for instance the Overconfidence Error(OE), Static Calibration Error (SCE), the Adaptive Calibration Error (ACE), and the Thresholded Adaptive Calibration Error (TACE), see [3] for reference.

- The use of inference time in Table 1 can be misleading, as this can vary based on the machine overload and low-level implementation details. It would be better to use the total number of Multiply-accumulate operations (MACs) instead.

References
----------

[1] Ashukha, A., Lyzhov, A., Molchanov, D., & Vetrov, D. (2020). Pitfalls of in-domain uncertainty estimation and ensembling in deep learning. arXiv preprint arXiv:2002.06470.

[2] Kumar, A., Liang, P., & Ma, T. (2019). Verified uncertainty calibration. arXiv preprint arXiv:1909.10155.

[3] Nixon, J., Dusenberry, M. W., Zhang, L., Jerfel, G., & Tran, D. (2019, June). Measuring Calibration in Deep Learning. In CVPR Workshops (Vol. 2, No. 7).

[4] Vaicenavicius, J., Widmann, D., Andersson, C., Lindsten, F., Roll, J., & Schön, T. (2019, April). Evaluating model calibration in classification. In The 22nd International Conference on Artificial Intelligence and Statistics (pp. 3459-3467). PMLR.

---

> ### Author Response · Authors · 2022-08-02
> **Response to Reviewer xEc2**
>
> Thank you for your insightful review and your encouraging words about our method and manuscript! Please find the individual answers to your raised points below:
>
> 1-The empirical evaluation has been performed on relatively small networks and simple datasets
> -------
> We do agree that testing further model architectures and datasets will make our evaluation more meaningful. However, we note that using relatively small networks and datasets is in line with other related papers studying fundamental questions about implicit ensembling  instead of practical applications. To address the relevant point of different proportions between number of layers and number of channels in more recent architectures, we have added numbers for the EfficientNet-B0 architecture.
>
> In general, extending the ideas to other architectures is not always straightforward, especially with limited time and computing resources (e.g., for Vision Transformers). However we would like to further explore the boundaries of our method in future works, including its applicability to diverse network architectures.
>
>
>
>
> 2-The Expected Calibration Error (ECE) being biased
> -------
> Thanks for that relevant insight! To reinforce our results regarding calibration, we have added the Brier score (as asked by reviewer 4hVr) as well as the Static Calibration Error (SCE). The latter should more accurately account for calibration by taking into consideration all classes, instead of just the one with the highest confidence.
>
>
>
>
> 3-Misleading use of inference times in Tab. 1
> -------
> We agree that inference time is not the only metric for the computational efficiency of algorithms, although it is of great relevance in practice. As suggested, we have added the total number of mult-adds to each architecture in Tab. 1. and conclude that most methods are approximately equal in that regard (~ 8.9 billion mult-adds for the ResNet-18 and ~2.45 billion for VGG-11). A notable exception is MIMO with 0.58 (0.18) billion mult-adds since it only requires a single forward pass, at the stark expense of model capacity per member.
>
>
>
>
> 4-Parameter initialization
> -------
> We use exactly the same initialization technique for a deep ensemble, which is the standard Xavier initialization. In addition, during the development, we observed that tuning the initialization gain factor in a deep ensemble does not affect the final performance, therefore we used the standard value (=1 in PyTorch).
>
>
>
>
> 5-Alternative ways to enforce/induce model diversity
> -------
> We would like to note that stochastic initialization and data sampling do not necessarily “enforce” model diversity, but rather naturally lead to functionally diverse, but equally well-performing optima in loss space. We did not apply any further means/techniques to diversify solutions, but your questions indeed points into a very interesting direction for future research.

---

> > ### Comment · Reviewer_xEc2 · 2022-08-07
> > **Thank you for the answer**
> >
> > Thank you for the answer, you have clarified most of my concerns. I will update the score from borderline to acceptance.
> >
> > I still think that an important missing part of the paper is the evaluation on larger models and datasets (concern shared by other reviewers).

---

> > > ### Author Response · Authors · 2022-08-09
> > > **response**
> > >
> > > Thank you for your constructive comments. As suggested we implemented FiLM for EfficientNet during the rebuttal period and we presented our results for EfficientNet-B0 (table below). We would like to run more experiments for a larger version of EfficientNet and include these results in the supplementary material. Regarding larger datasets, we would like to repeat that we would like to add an ImageNet comparison for the final camera-ready version if time allows.

---

### Official Review · Reviewer_uuAJ · 2022-07-11

**Rating:** 6
**Confidence:** 4
**Soundness:** 3 good
**Presentation:** 3 good
**Contribution:** 3 good

**Summary:**

This paper proposes to use feature wise linear modulation [1] to construct implicit deep ensembles. In FilM [1], the authors propose to replace batch norm layer in a deep architecture with a conditional batch norm layer, with two parameters (shifting and scaling coefficients), which are used to transform features given a conditional input. In this work, the authors propose to use film layer, but to use multiple sets of affine parameters to instantiate different ensemble members.
The authors show for a couple of datasets (cifar10/100, retina glaucoma detection, not imagenet), some networks have performance better than other implicit ensemble approaches with respect to deep ensemble performance both in terms of accuracy and expected calibration error.

Refsparame
[1] Perez et al. 2018. FilM

**Questions:**

- How easy is it to learn the hyperparameters of film ensembles?
- Have the authors tried to reproduce their results on larger datasets, i.e. imagenet?
- How do the authors expect this approach to perform in tasks where fine tuning/using pretrained networks is necessary?


**Limitations:**

Yes

**Strengths And Weaknesses:**

Strengths:
- The paper is well written, where the authors explicitly state the contributions, comparison metrics and results.
- The authors compare their proposed implicit ensemble model to a wide variety of existing implicit ensemble models.
Weaknesses:
- The paper would greatly benefit from a discussion on how the proposed film ensemble is different from MIMO or batch ensemble (BE) approaches. These could be added with minor equations in the appendix. Currently, the authors only state in line 163 "[BE]... is closest to our work".
- From the paper, it is unclear to me how difficult it is to learn the film parameters for this model. In my experience, finding the best hyper parameters for batch ensemble models requires min hyper search, is this also true here?
-  The authors mentioned that we can expect to trade-off accuracy vs gain given $\rho$ which controls the variance of the weight initialization. From Fig 3, it is unclear if there is a pareto optimal solution (tradeoff between accuracy and ece) or fig 3 is simply an artifact for a particular dataset. Adding additional fig for other datasets would be helpful.
- The authors state some failure modes for other implicit ensemble approaches, i.e. "batch ensemble generally exhibits a negative correlation between ensemble size and test set accuracy", "mimo shows very poor performance.." These results to my knowledge are not aligned with published work. For this part I would suggest to for the authors to better discuss the limitations of their baseline implementations. Conversely, if the authors do believe that this a standard result/common knowledge, they can explicitly state so/cite similar findings. This could be in addition to the discussion of how difficult it is to optimize the hyperparameters for film/other methods to construct implicit ensembles.
- The datasets considered are low dimensional/ low resolution. Have the authors tried to reproduce their results on larger datasets, i.e. imagenet?

---

> ### Author Response · Authors · 2022-08-02
> **Response to Reviewer uuAJ**
>
> Thank you for the detailed review! We addressed all points below:
>
> 1-How is the proposed film ensemble different from MIMO or batch ensemble (BE)?
> -----
>
> Although all mentioned methods are instances of implicit ensembling methods that try to emulate a model ensemble while being less resource-demanding, there are important differences between them. FiLM-Ensemble works by learning diverse, layer-wise affine (“feature warping”) transformations for every ensemble member, simulating a deep ensemble with significantly less time- and storage complexity. BE try to achieve this effect by learning a member-wise multiplicative low-rank matrix (that can be stored efficiently) for each weight matrix of the network. Finally, MIMO operates fundamentally differently, making multiple predictions simultaneously in one forward pass, effectively dividing the original model’s capacity by M.
>
> As suggested, we will add equations to the supplementary material and also try to detail the differences better in the final version of our manuscript.
>
>
>
> 2-The datasets considered are low dimensional/ low resolution. Have the authors tried to reproduce their results on larger datasets, i.e. imagenet?
> -----
>
>
>
> The REFUGE2020 dataset that was used in our experimental evaluation has a resolution of 1411x1411 pixels, making it considerably higher-resolved than ImageNet. However, as stated in the responses to other reviewers, if time allows, we would like to add a comparison on ImageNet for the final version of the paper.
>
>
>
>
>
> 3-How hard is it to find good parameters for FiLM?
> -----
>
> There is only one hyperparameter introduced by our method: gain, which is discussed in Section 3.6. Nevertheless, we set this hyperparameter to a certain value (which is 2) for all our experiments and did not have to further optimize this hyperparameter except in Section 3.6, where we analyze this very hyperparameter. As shown in this section, increasing the value of gain increases model calibration in terms of ECE at the expense of a minor accuracy drop. In short, it is very simple to find good hyperparameters for FiLM-Ensemble and the method achieves high performance without optimizing any hyperparameters whatsoever (for example by using the default gain of 2).
>
>
>
>
>
>
> 4-How do the authors expect this approach to perform in tasks where fine tuning/using pretrained networks is necessary?
> -----
>
> We recognize the practical relevance of this question and would like to leave this as an open question for successor works. The diversification of ensemble members takes place by means of learning independent affine feature transformations, and “ordinary” network parameters are shared between members anyway. Therefore, we can imagine our approach to compare favourably against ensembling methods that rely on the diversification of the network parameters themselves. However, this is pure speculation that remains to be tested thoroughly in future work.
>
>
> 5-Failure modes for other implicit ensemble approaches
> -----
>
> We will try our best to add further relevant references to the camera-ready version. All claims that are being made in the paper are backed by our evaluations, for which source code will be publicly available (including our implementations for all comparison methods). As we have wrote in our reply to reviewer #4hVr, we will further add a dedicated limitations section to the final manuscript. We are happy to include possible comparison method failure modes as well.

---

> > ### Comment · Reviewer_uuAJ · 2022-08-09
> > **response**
> >
> > Thanks to the authors for their response. In particular, I appreciate the willingness of the authors to note in detail additional metrics and the limitations they encountered while reproducing results for other implicit ensemble models, re MIMO, BE.
> > I really like the simplicity of the proposed approach. However, as a reviewer, for me to really champion the paper and update my score to a 7 or even 8, I would like to see the following changes:
> > (1) Higher dimensional comparisons: While the REGUGE2020 dataset is high dimensional 1411x1411, my understanding is that the version of the data the authors are using is for a binary task, not a segmentation/multiclass class task.
> > (2) The authors did not comment if the results for Fig 3 were dataset dependent or if we can find a pareto optimal solution (tradeoff between accuracy and ece) across datasets.
> > (3) A more formal description of why mode collapse does not occur in their approach.
> > (4) An evaluation for pretrained models. As we know, pretrained architectures are pervasive, so including such an analysis would greatly increase the impact of the paper in my opinion.

---

> > > ### Author Response · Authors · 2022-08-09
> > > **response**
> > >
> > > Thank you for your constructive comments and please find the individual answers to your raised points below:
> > >
> > >
> > > 1-Higher dimensional comparisons: While the REGUGE2020 dataset is high dimensional 1411x1411, my understanding is that the version of the data the authors are using is for a binary task, not a segmentation/multiclass class task.
> > > ----
> > > Yes, REGUGE2020 is a very high dimensional problem while it is a binary classification task but we are not sure why it is a concern - it is a very important medical task and it is a real-world dataset.
> > >
> > >
> > > 2-The authors did not comment if the results for Fig 3 were dataset dependent or if we can find a pareto optimal solution (tradeoff between accuracy and ece) across datasets.
> > > ----
> > > Thank you for pointing this out, we had overlooked this comment.  It is not dataset specific and we will add the same figure for another dataset in the supplementary material. Here we present numbers for Cifar100/Resnet34 (e=2).
> > >
> > > Resnet34/Cifar100 (e=2)
> > > |Gain |Acc |ECE
> > > |---|---|---|
> > > 2| **80.47**|0.0580
> > > 5|79.76|0.0527
> > > 10|79.72|0.0425
> > > 25|79.39|**0.0398**
> > >
> > >
> > >
> > >
> > >
> > > 3-A more formal description of why mode collapse does not occur in their approach.
> > > ----
> > > Unfortunately, we are not sure what the reviewer means, we think that it is about the failure modes of baseline methods. We would like to elaborate our point. As we already mentioned in the text these methods' effectiveness is limited to wide and large architectures and quite sensitive to hyper-parameter choice. When you decrease the network capacity or increase the number of ensembles they are not effective at all. This is also in line with their experiments where they only show results with large architectures e.g Resnet-28-10 and only 2 or 3 ensemble members. In addition, in our direct contact with the authors of MIMO, it is confirmed that to replicate the numbers, all the hyperparameters have to be exactly matched (even the number of epochs and large batch size and so on), which have high compute requirements. In addition, we will release all the baseline implementations.
> > >
> > >
> > >
> > >
> > > 4-An evaluation for pretrained models. As we know, pretrained architectures are pervasive, so including such an analysis would greatly increase the impact of the paper in my opinion.
> > > ----
> > > As we mentioned before, this is not a straightforward analysis because we change the architecture of the network; thus it requires initializing the network parameters from a different network. This is an issue not only for the proposed approach but for all the compared approaches e.g Batch-Ensemble or MIMO. In addition, this is not discussed in any of the related papers But we think that your question indeed points in a very interesting direction for future research.

---

### Official Review · Reviewer_gCqT · 2022-07-12

**Rating:** 3
**Confidence:** 3
**Soundness:** 2 fair
**Presentation:** 3 good
**Contribution:** 3 good

**Summary:**

This paper proposes a simple implicit model ensemble method based on feature-wise linear modulation (FiLM), where they learn M sets of affine parameters for batch norm layers. Experiments show higher diversity than DE, high accuracy and ECE among implicit ensemble methods, and good OOD detection capabilities.

**Questions:**

In table 1, what causes the difference in inference time between MIMO and FiLM-Ensemble for ResNet-18? Aren’t both methods similar to feeding forward a batch of 16 datapoints (in terms of computation cost)?

Why do other methods perform so poorly in Table 2? If these methods were adequately tuned, could you report results on CIFAR10/100 with ResNet-28-10 and compare them to the results in the MIMO paper?

**Limitations:**

To my knowledge, this work has no potential negative societal impact other than what was already present in the existing literature.

**Strengths And Weaknesses:**

The proposed method is conceptually simple in addition to being parallelizable.

The M sets of affine parameters are learned concurrently, and the M ensemble members would collapse to the same function if these parameters were the same. Do you have an intuition for why this does not happen? In other words, why is the different random initialization sufficient for diversity at the end of training, given that the functions share all other parameters?

I don’t understand why Figure 1 happens; why should the pairwise diversity of FiLM-Ensemble be bigger than that of DeepEns if nothing in the objective is directly encouraging diversification? If anything, it seems like it should be lower because of the shared parameters. Also, why does diversity increase with the number of members if these metrics are averages of pairwise metrics? As you add more models, it intuitively seems that some of them should “overlap” and thus have lower diversity.

Table 2 is unconvincing to me. The other networks (MIMO, BE, etc.) are doing worse than a single network, making it seem like the hyperparameters were not adequately tuned for these methods. For example, MIMO and BE perform better than a single deterministic network in the MIMO paper.

Line 107: “…depend bot on” typo
Line 116: “thid” typo

---

> ### Author Response · Authors · 2022-08-02
> **Response to Reviewer gCqT**
>
> Thank you for your review and your constructive criticism! We have tried to adequately answer your questions below:
>
> 1-Why don’t ensemble members collapse into a single solution?
> -----
> We agree that this is a very interesting question, which equally arises for other implicit ensembling methods. In our case, our intuition is that affine transformations are generally extremely powerful operations (e.g., they can fully invert features or dramatically change the scaling between them). This gives rise to a very complex and high-dimensional loss landscape (even when “ordinary” model parameters are fixed), making it almost impossible for ensemble members to converge to the same local optima when starting from random initializations. Still, we agree that this is a question that deserves further, more formal, and systematic investigation in future works, not only for the specific case of our feature-wise affine modulations but for low-rank implicit ensembling methods in general. However, we think this systematic investigation is out of the scope of our paper.
>
>
>
> 2-Diversity of FiLM-Ensemble vs. Deep Ensemble in Fig.1
> -----
> The question of why FiLM-Ensemble exhibits higher inter-member diversity than deep ensembles naturally arises from Fig. 1, we will add a sentence to the manuscript explaining our intuition for that effect. Firstly it is important to note that deep ensembles are not encouraged during training to be diverse either, and that their diversification is merely a consequence of stochastic initialization and training on a highly complex, multi-modal loss surface. We believe this is true for the affine FiLM parameters as well, even when the “ordinary” parameters are fixed, see the previous answer. Since the relevant FiLM parameters are more sensitive, we believe such random initialization and stochastic training induce higher variability in the resulting submodels.
>
>
>
> 3-Why are other pseudo-ensemble methods worse than a single network?
> -----
> Most models reduce the capacity of individual members (e.g., MC dropout, most notably MIMO), but expect the ensemble to compensate for that. Our method did not suffer from this effect in our evaluations.
>
>
>
> 4-For example, MIMO and BE perform better than a single deterministic network in the MIMO paper. Why do other methods perform so poorly in Table 2? If these methods were adequately tuned, could you report results on CIFAR10/100 with ResNet-28-10 and compare them to the results in the MIMO paper?
> -----
> We already tried to convey in the text that these methods' effectiveness is limited to wide and large architectures and quite sensitive to hyper-parameter choice. When you decrease the network capacity or increase the number of ensembles they are not effective at all. This is also in line with their experiments where they only show results with large architectures e.g Resnet-28-10 and only 2 or 3 ensemble members. In addition, in our direct contact with the authors of MIMO, it is confirmed that to replicate the numbers, all the hyperparameters have to be exactly matched (even the number of epochs and large batch size and so on), which have high compute requirements.
>
>
>
> 5-Why is MIMO so much faster than FiLM for ResNet-18?
> -----
> MIMO requires a single forward pass rather than M forward pass. This implies that the network is expected to contain M subnetworks, which is why the original paper focuses on wide ResNets to retain sufficient capacity for the submodels. When using ResNet-50, the authors only explore M=2 for this reason. The original paper shows that even for the wide ResNet-28-10, the performance already drops for M > 2, suggesting that the network does not have enough spare capacity to simulate that many subnetworks at once. A direct comparison between MIMO and other methods is not trivial since the backbone would need to be widened somehow, which would counterbalance its apparent efficiency advantage.

---

> > ### Comment · Reviewer_gCqT · 2022-08-09
> > **Response to Authors**
> >
> > First, I thank the authors for their response. After reading the response and other reviews, I would like to maintain my score for the reasons below.
> >
> > Points (1, 2, 5) all make sense to me, and I agree: studying why ensembles do not collapse is out of scope for this paper, FiLM parameters may be especially sensitive, and I now understand the compute requirements of MIMO.
> >
> > Regarding my concerns about evaluations (points 3 and 4), I still think the evaluation is incomplete. Table 2 essentially claims the opposite of the existing literature, i.e., all previous methods perform worse than even a single network. I understand from the rebuttal that the differences may be because of architecture and hyperparameter choice. Still, I think there needs to be more of an effort to reconcile the differences with the existing literature by evaluating FiLM-Ensemble in the existing setting (ResNet-28-10 etc) and/or thoroughly showing the results of a hyperparameter sweep in the newer setting (ResNet-18 etc). Additionally, given that the evaluation setting used in the paper uses a smaller network with lower performance compared to the setting in previous works, I think this choice should be justified further.

---

> > > ### Author Response · Authors · 2022-08-10
> > > **Response**
> > >
> > > Thank you for your constructive comments and please find the individual answers to your raised points below:
> > >
> > >
> > > **This is not true:**  *Table 2 essentially claims the opposite of the existing literature, i.e., all previous methods perform worse than even a single network.*
> > >
> > > Please see some results from existing literature below.
> > >
> > > Table 1 in [1]: Cifar10/WideResnet28-10
> > >
> > > | |Acc|cAcc |ECE |NLL
> > > |---|---|---|---|---|
> > > Single|**96**|**76.1** | **0.023** | **0.159**
> > > MC-Drop|95.9	| 68.8| 0.024|0.160
> > >
> > >
> > >
> > > -------------------------------------------
> > > Table 3 in [1]: Cifar100/WideResnet50
> > >
> > > | |Acc|cAcc |ECE |NLL
> > > |---|---|---|---|---|
> > > Single |	76.100|40.500	|   **0.039** |	 **0.943**
> > > BatchEnsemble|**76.700**|**41.800**| 0.049| 0.944
> > >
> > >
> > >
> > > -------------------------------------------
> > >
> > > Table 2 in [2]: Perplexity on Newstest2013 with big Transformer. BatchEnsemble with ensemble size 4.
> > > |  |Perp|
> > > |---|---|
> > > |Single |2.76|
> > > |MC-Drop |  2.77|
> > > |Batch-Ensemble| **2.74**|
> > >
> > >
> > >
> > >
> > > -------------------------------------------
> > >
> > >
> > > Table 2 in [3]: ImageNet/Resnet50 results.
> > >
> > > |			|	Acc	|	ECE
> > > |---|---|---|
> > > Single 			|	**0.71** |	0.07
> > > MC-Drop		|	0.69	|	0.03
> > > Masksemble		|	0.70	|	 **0.02**
> > >
> > >
> > > As you see in the tables single method can work better than other implicit methods in many different experimental settings. Those implicit ensemble methods usually improve calibration performance at the cost of accuracy and sometimes they do not work at all, see the MC-Dropout performance in the first table.
> > >
> > >
> > > **This is not true:** *all previous methods perform worse than even a single network.*
> > >
> > > Table 2 shows that Deep Ensemble consistently works best in terms of accuracy and calibration and the proposed method comes second with a small margin. Table 2 also shows that when we increase the capacity of the network (Resnet18→Resnet34), other implicit methods except MIMO improves the calibration performance at the cost of accuracy. This is also aligned with the existing literature, please see the tables above.
> > >
> > >
> > >
> > >
> > >
> > >
> > > I think there needs to be more of an effort to reconcile the differences with the existing literature by evaluating FiLM-Ensemble in the existing setting (ResNet-28-10 etc) and/or thoroughly showing the results of a hyperparameter sweep in the newer setting (ResNet-18 etc).
> > > ---
> > > As we mentioned before, we did our best effort to implement these baseline methods, and quite a few colleagues independently confirmed these implementations - as we promised we will release these baseline implementations. We even contacted via direct email the authors of MIMO and Masksemble for confirmation of our implementations but we did not get a reply regarding our implementations. We only get this information: to replicate the numbers of MIMO, all the hyperparameters have to be exactly matched (even the number of epochs and large batch size and so on), which have high compute requirements.
> > >
> > >
> > > Also, we present results in a much more diverse setting than baseline methods’ papers: different combinations of 5 different datasets={Cifar10, Cifar100, Glaucoma, REFUGE, 6mA}, 6 different architectures={VGG11, Resnet18, Resnet34, EfficientNet-B0, 1-dim CNN, ViT}, and 4 different numbers of ensembles={2,4,8,16}) which is already much more than compared methods, e.g MIMO only shows results on 3 different datasets, 2 architectures and 2 different numbers of ensembles={2,3}.
> > >
> > > We are unsure why we are required to choose exactly the same experimental setting with baseline methods (also each of them uses a different setting) unless it is a benchmark. For instance, in the MIMO setting, they use ResNet28-10 for Cifar10 with a batch size of 512 (see Appendix B in [1]). We have  NVIDIA GeForce GTX 1080 Ti and in this setting, we encounter a memory error.
> > >
> > > Last but not least, some previous methods find a setting to make the algorithm work with high compute requirements which is already opposed to the idea of finding efficient ensemble methods i.e losing the main idea. Please see Section 6: Broader Impact in our paper. The main idea is to improve performance while reducing compute costs. If some algorithms are very fragile with hyperparameters, then they are actually useless for society/community - unfortunately, not everybody has the time/money to find those specific settings.
> > >
> > >
> > >
> > >
> > >
> > > [1] Havasi, Marton, et al. "Training independent subnetworks for robust prediction." ICLR, 2021.
> > >
> > > [2] Wen, Yeming, Dustin Tran, and Jimmy Ba. "Batchensemble: an alternative approach to efficient ensemble and lifelong learning." ICLR, 2020
> > >
> > > [3] Durasov, Nikita, et al. "Masksembles for uncertainty estimation." CVPR, 2021.

---

### Author Response · Authors · 2022-08-02
**Response to all reviewers and AC**

We would like to thank all reviewers for their constructive comments and criticism which we have tried to address accordingly (see individual comments for detailed responses). We are happy that our paper was received well and that the proposed method is considered novel, simple and effective.

The significant points that we further clarified in response to the reviews are:

1-We clarified some open questions w.r.t. Fig. 1 and Tab. 2 and the overall convergence behavior of our method.

2-We added two more metrics (SCE, and Brier score) to reinforce our claims about calibration.

3-We implemented our method with a more recent model architecture: EfficientNet-B0 to show our method can be easily and effectively adapted to new architectures.

4-We added missing error bars to tables and figures in the manuscript.

5-We added 3 state-of-the-art OOD detection baselines to Table 5 in the manuscript to more highlight our method's effectiveness for OOD detection.

6-We added 3 state-of-the-art baselines to Retinal Glaucoma Detection (Section 3.4, Table 4 in the manuscript) to strengthen our comparisons.

7-We added the total number of multiplication-addition operations to Tab. 1 in the manuscript.

8-Fixed minor issues in the paper that were brought to attention by any of the reviews (e.g., spelling mistakes).

---

### Meta-Review · Area_Chair_DS5P · 2022-08-27

**Recommendation:** Accept
**Confidence:** Less certain

**Metareview:**

This paper constructs implicit ensembles by adding multiple affine transformations to batch normalization layers. The proposed approach is elegant and complements existing implicit ensemble techniques. There are some concerns that the experimental evaluation does not include larger models or datasets - which is important because other implicit ensemble models (e.g. MIMO) exhibit sensitivity to model/dataset size. It would also be useful to discuss how this method could be used in conjunction with pre-trained models. Nevertheless, the idea is well-presented, simple, and effective, and therefore it will be of interest to the NeurIPS community.

**Award:**

No

---

### Decision · Program_Chairs · 2022-09-14

Accept